# The Role of Environmental Regulation and Technological Innovation in the Employment of Manufacturing Enterprises: Evidence from China

**Die Li * and Jinsheng Zhu**

School of Economics, Wuhan University of Technology, Wuhan 430070, China; zhujs@whut.edu.cn
* Correspondence: lidie1004@whut.edu.cn

**Abstract:** With the increasingly severe emission reduction pressures, it is an inevitable choice for China to improve the intensity of environmental regulation. At the same time, the impact of technological innovation on enterprise employment may lead to some new changes under the environmental regulation constraints. However, existing studies have not included environmental regulation into the theoretical framework of technological innovation and enterprise employment, nor has the influencing mechanism of environmental regulation and technological innovation in the employment of manufacturing enterprises been explored. This paper uses the panel data of listed manufacturing companies in the A-share market of Shanghai and Shenzhen from 2011 to 2017 to examine the impact of environmental regulation and technological innovation on the employment of manufacturing enterprises, and explore their influence mechanisms in a theoretical framework based the moderating effect model. The findings demonstrate the following: First, the technological innovation has a positive creative effect on enterprise employment. Second, the impact of environmental regulation on enterprise employment is significantly positive. Third, environmental regulation has a negative moderating effect on the impact of technological innovation on enterprise employment. Finally, the impacts of both environmental regulation and technological innovation on the employment of manufacturing enterprises are heterogeneous across enterprises due to differences in ownership structure, the degree of pollution, and technical density. Therefore, faced with the objective reality that environmental carrying capacity has reached or approached the upper limit, China needs to formulate a differentiated and diversified technological innovation system and environmental protection policy, improve the environmental innovation level of manufacturing enterprises, and form a green development model, which is of great significance for achieving high-quality development and stable employment.

**Keywords:** environmental regulation; technological innovation; employment of manufacturing enterprises; enterprise heterogeneity

## 1. Introduction

For a long time, China has relied on the comparative advantage of lower labor costs and environmental costs to participate in the international division of labor, promote the expansion of production scale through continuous investment, and realize the rapid growth of economic aggregation. With the increasingly strengthened domestic environmental constraints, diminishing marginal benefits of capital accumulation, and the rising labor costs, it is difficult to support the continued rapid growth of the economy. Internationally, China's manufacturing industry is also facing the low-end lock of the global value chain dominated by Europe and the United States. In the context of internal and external troubles, shifting economic growth to relying on total factor productivity and achieving

innovative growth is the inevitable way out for the current Chinese economy. In the 2019 government work report (the government work report is a report delivered by the premier of the state council to the National People's Congress and submitted to deputies of the National People's Congress for deliberation) of the two sessions (NPC and CPPCC), the employment priority was raised to the level of national macro-control for the first time. Stabilizing employment is not only promoting people's livelihood but has become an important driving force for the country's high-quality development and the transformation of old and new growth drivers. It is necessary to rely on innovation in manufacturing to bring about high-quality economic development, and to solve the employment problem to reduce the social burden of employment, so as to achieve the dual goals of high-quality development and full employment.

The fourth industrial revolution is springing up and bringing about numerous new industries and new economic forms, which has been called the "innovation economy." On the one hand, the innovation economy can create a large number of new jobs. On the other hand, the innovation economy has a squeezing effect on employment and brings unemployment shocks [1]. In recent years, the rapid development of new technologies, such as artificial intelligence and robots, not only provides new impetus for economic growth, but also triggers the panic of "machines replacing people." At present, the destructive effect of technological progress in artificial intelligence and robots on employment is limited, but the long-term employment effect is not optimistic [2]. Some scholars are concerned about how susceptible current jobs are to these technological developments [3], while others have explored the employment creation effects of innovation activities [4,5]. Moreover, relevant literatures have further studied the relationship between environmental innovation and employment creation. Horbach and Rennings [6] examined the employment effects of environmental technologies in different fields of environmental innovation, and the empirical results show that the introduction of cleaner technologies as process innovations leads to a higher employment by improving the competitiveness of firms, but air and water process innovations that are still dominated by end of pipe technologies have a negative impact on employment. Gagliardi et al. [7] investigated the link between environment-related innovation and job creation at the firm level. The econometric analysis shows a strong positive impact of "green" innovation (measured by the number of environment-related patents) on long-run job creation, which was substantially bigger than the effect of other innovations. Triguero et al. [8] discussed the synergistic effect between eco-innovation and employment based on a sample of more than 6000 innovative Spanish manufacturing and service firms. The main findings show that size, research and development (R&D), and export influence eco-innovation and employment in the same direction. To sum up, more and more studies attempt to explore the potential employment effect of environmental innovation but fail to reveal the impact path of environmental regulation on technological innovation and enterprise employment.

The controversial debate on the relationship between environmental regulation and technological innovation has been going on for a long time, and most of the literature validates the innovation compensation effect of environmental regulation. Porter hypothesis holds that strict and appropriate environmental regulation can stimulate enterprises' innovation, partially or even completely offset the cost of enterprises' compliance with environmental regulation, and improve enterprises' international competitiveness [9]. On the premise of pursuing profit maximization, environmental regulation imposes additional constraints on enterprises, and enterprises may change their original behaviors and carry out innovative activities so as to reduce costs under the new constraints. Jaffe and Palmer [10] examined the impact of environmental regulation on R&D expenditure and patent application volume by using data from the U.S. industrial sector and found that environmental regulation significantly promoted R&D expenditure but had no significant impact on patent application. Some scholars found a positive relationship between environmental regulation and patent application through empirical research on the number of environment-related patent applications [11,12]. Domestic scholars have found that there is a U-shaped relationship between environmental regulation intensity and technological innovation, and it can only be realized when environmental regulation intensity crosses a certain threshold

value [13,14]. Milani [15] found that industries that are not easily transferred will carry out more research and development activities in the face of stronger environmental regulation as an alternative to industrial relocation. It can be seen that there is no consistent conclusion regarding how environmental regulation affects technological innovation in academia, and the impact of environmental regulation on technological innovation may be related to industry characteristics.

The academic community agrees that there are two mechanisms through which environmental regulation impacts on employment: negative scale effect and positive substitution effect. The early studies mainly focused on the scale effect and argued that environmental regulation would lead to the increase of production cost and governance cost of enterprises, weaken the competitive advantage of enterprises, and then lead to the reduction of enterprise scale and the reduction of labor demand [16]. However, when environmental regulation raises the price of resource production factors, enterprises' productive input tends to be labor-intensive, leading to the increase of labor input factors, and thus producing a substitution effect [17]. Therefore, the employment effect of environmental regulation depends on the size of the scale and substitution effects. With the deepening of research, scholars have found that the impact of environmental regulation on employment presents different characteristics in different countries, regions, and industries. Many empirical studies show that, from the national or local level, differences in environmental regulation standards will lead to international and regional industrial transfer, resulting in the spatial transfer of employment and uncertainty of the impact of environmental regulation on employment [18]. From the perspective of industry, the impact of environmental regulation on employment is heterogeneous among industries, which will lead to the flow of labor among industries, resulting in the linkage between various industries [19].

With the increasingly severe emission reduction pressures, it is an inevitable requirement to improve the intensity of environmental regulation before reaching the environmental carrying capacity. According to Porter hypothesis, appropriate environmental regulation may stimulate technological innovation of enterprises, while the enterprises' change of production technology will produce an uncertain employment effect. From the perspective of the stage of technology adoption by enterprises, the process of enterprises' improvement or introduction of advanced clean production technology to obtain technological progress forces enterprises to crowd out production and investment, which will affect the scale and market share of enterprises, thus adversely affecting employment. With the application of cleaner production technology, costs begin to be offset or even recovered, and the demand for environmentally sound products increases. Enterprises that take the initiative to adopt cleaner production technology will gain a higher market share and provide more jobs. From the perspective of the types of technologies adopted by enterprises, if enterprises choose production-oriented technological progress, it may produce a crowding out effect on the labor force due to the improvement of capital intensity. If enterprises choose to make progress in pollution control technology, it will promote the development of the environmental protection industry and create new labor demand. What is the overall effect of technological innovation on employment growth under environmental constraints? Is environmental regulation holding back the creation of jobs through technological innovation? The answers to these questions will help us better understand and resolve the dilemma of high-quality development and employment growth under current environmental regulation.

While the existing studies do lay the foundation and offer some inspiration for this paper, our study is one of the first that explores the regulatory role of environmental regulation in the impact of technological innovation on enterprise employment in China. Based on the direct impact of technological innovation on enterprise employment, as a new perspective, this paper aims to take environmental regulation as a moderating variable to analyze the relationship between technological innovation and enterprise employment, which will make the research on the relationship between technological innovation and employment more accurate and comprehensive. The marginal contribution of this paper is as follows: First, this paper brings environmental regulation, technological innovation, and enterprise employment into the same analytical framework in order to identify the mechanism of environmental regulation on the complex relationship between technological innovation

and employment growth. Second, this paper is the first to adopt the moderating effect model to investigate how environmental regulations affect the relationship between technological innovation and enterprise employment based on the A-share manufacturing companies listed in Shanghai and Shenzhen stock exchanges from 2011 to 2017. Third, this paper explores the differences in the impacts of environmental regulation and technological innovation on the employment of manufacturing enterprises for different types of enterprises. The main purpose of this study is to distinguish the impact of environmental regulation of enterprises with different ownership structures and different industry characteristics on the relationship between technological innovation and employment so as to provide a reference for the government to formulate effective environmental policies and innovation policies for a range of enterprises.

This paper is organized as follows. Section 2 provides an overview of the related literature and proposes the research hypotheses. Section 3 introduces the design and methodology of this study and shows the variables chosen as well as data sources. Section 4 describes our empirical results and presents our discussion. Finally, Section 5 provides conclusions and policy implications.

## 2. Literature Review and Research Hypotheses

### 2.1. Technological Innovation and Enterprise Employment

The impact of technological innovation on employment can be traced back to the dialectical study of structural unemployment by the British classical economist Ricardo. Katsoulacos [20] looked at the employment effect of product innovation in a general equilibrium setting, seeking to obtain theoretical support for the claim that product innovation leads to an increase in the equilibrium level of employment. Freeman and Soete [21] discussed the impact of computerized technical change on employment in the 21st century. Vivarelli [22] addressed the impact of technical change on employment from both theoretical and empirical perspectives. The theoretical discussion and empirical results were combined to demonstrate that the employment impact of labor-saving technologies can only be partially counter-balanced by market forces, and therefore economic policy measures may be necessary. From a micro-economic perspective, the impact of technological innovation on the number of jobs in enterprises is analyzed. Most scholars argued that technological innovation contributes to employment growth [23–26]. Relevant research used the employment equation similar to the labor demand formula to study the impact of technological innovation on total employment or analyzed the impact of technological change on the employment creation rate and destruction rate. Hall et al. [27] empirically analyzed the data of Italian manufacturing enterprises with the extended HJMP model (a model framework based on production functions). The results indicated that technological innovation had a positive effect on employment growth and enterprise productivity, while product innovation and old product sales growth contributed to half of the employment growth. Based on the panel data of German manufacturing enterprises, and using the dynamic GMM model to test the impact of technological innovation on the employment, Lachenmaier and Rottmann [28] found that technological innovation has a positive effect on employment and a lag effect, and that process innovation has a greater effect on employment than product innovation. The research on the impact of technological innovation on employment is relatively late in China. Zhu and Li [29] used the data of China's large and medium-sized industrial enterprises to analyze the total effect of technological progress based on the stochastic frontier method and proposed that the increase of employment brought by technological progress can compensate for the loss of employment impact. Huang et al. [30] investigated the horizontal and hysteresis effects of technological innovation on employment by using panel data of listed manufacturing companies in China. Their simulation results show that technological innovation is negatively related to employment level, but technological innovation has a significant positive hysteresis effect on employment growth, which means technological innovation contributes to employment growth in the long run. Wu [31] investigated the employment creation effect of different types of innovation in enterprises and found that technological innovation had a significant

effect on employment promotion. Some scholars have obtained different results. Based on the data of large and medium-sized manufacturing enterprises in China, He and Qian [32] studied the impact of technological innovation on the survival and employment growth of enterprises. The research results showed that technological innovation had a positive effect on the survival of enterprises but had no effect on employment growth. Han et al. [33] empirically examined the impact of technological innovation on employment growth from the perspective of process innovation, product innovation, and enterprise research and development by using the survey data of Chinese industrial enterprises in the World Bank database in 2012, and found that technological innovation is negatively correlated with employment growth. Existing studies have compared and examined the "creation effect" and "destruction effect" of technological innovation on employment under different characteristic scenarios, but fail to consider the possible evolution of the relationship between technological innovation and employment under the background of environmental regulation. Furthermore, the employment effect of enterprises may be related to factors such as the demand for technological innovation under the pressure of environmental regulation.

Many scholars have further analyzed the heterogeneity impact on employment of technological innovation of enterprises in different industries. Bogliacino and Pianta [34] investigated the relationship between innovation and employment through a model and empirical test at industry level for eight European countries for 1994–2004 and proposed a revised Pavitt taxonomy in order to identify specific patterns of technological change and job creation and loss. Bogliacino et al. [35] tested the job creation effect of business R&D applying the dynamic LSDVC estimator to a longitudinal database covering 677 European companies over the period 1990–2008. They found that job creation was detected in services and high-tech manufacturing, but not in traditional sectors. Bogliacino and Vivarelli [36] used a unique database covering 25 manufacturing and service sectors for 15 European countries over the period 1996–2005, for a total of 2295 observations, and applied GMM-SYS panel estimations of a demand-for-labor equation augmented with technology. They found that R&D expenditure (fostering product innovation) has a job-creating effect, in accordance with the previous theoretical and empirical literature discussed in their paper. Evangelista and Vezzani [37] explored the employment impact of innovation extending the analysis to organizational change based on the firm-level data provided by the fourth Community Innovation Survey (CIS4), and the empirical results show that both technological and organizational innovation exert a positive impact on employment mainly "indirectly," that is by improving growth performances in firms. By using a dataset made of 879 large international firms observed for the period 2002–2010 and localized in USA, Japan, and Europe, Aldieri and Vinci [38] analyzed the extent to which the economic crisis may affect the sensitivity of employment with respect to their own innovation but also with respect to outside innovation, i.e., the R&D spillovers, in high-tech and low-tech industries. Falk and Hagsten [39] investigated the impact of new market product (market novelty) sales on labor demand (employment) by using biennial data for 25 industries, nine European countries, and five time periods (2002–2010), and the GMM estimations show that the turnover (sales) of market novelties (in relation to existing products) has a significant impact on relative employment in manufacturing industries, while employment in service industries does not benefit from new market products but instead from the intensity with which information and communication technology innovations are used. Piva and Vivarelli [40] discussed the economic insights on the employment impact of technological change covering both classical theories and updated theoretical and empirical analyses, and provided an empirical test based on longitudinal data covering manufacturing and service sectors over the 1998–2011 period for 11 European countries. The main results show that a significant labor-friendly impact of R&D expenditures (mainly related to product innovation) was found; yet, this positive employment effect appears to be entirely due to medium- and high-tech sectors, while no effect was detected in low-tech industries.

The employment loss effect of technological innovation mainly comes in two forms. First, the increase in labor productivity brought by technological innovation will cause short-term unemployment. The improvement of production tools, production technology, and production objects by technological

innovation not only reduces labor intensity but also largely replaces part of the labor force. Enterprises will determine the actual demand of the labor force based on marginal productivity, which undoubtedly directly reduces employment. Of course, the production demand of different industries and products is different, and the organization of production factors is different; moreover, the labor demand reduced is also different. Technological innovation can also improve the productivity of labor by promoting the optimization of management organization, streamlining management processes, improving production efficiency, and strengthening labor training so as to replace the ordinary workers who are not matched with labor skills and jobs with a small number of high-skilled labor force and reduce the employment of labor force. The second form is that technological innovation leads to a decline in labor demand by increasing capital productivity and forming an alternative effect of factors of production. Technological advances triggered by technological innovation can change specific combinations of factors of production, thereby changing the proportion of capital and labor. With the development of technology and the improvement of the quality of innovation, capital-biased technological innovation is becoming more and more common, that is, the relative marginal output of capital increases more than the relative marginal output of labor. From the perspective of industrial development, labor-intensive industries are facing a situation of being gradually replaced by capital, knowledge, and technology-intensive industries. The replacement of new and old industries has gradually reduced the demand for labor.

The employment creation effect of technological innovation is the social effect of expanding labor demand by creating and increasing employment opportunities. First, technological innovation can further absorb more labor force by increasing investment and expanding production scale. Technological innovation brings about the increase of effective demand, and enterprises get rich profits. Under the condition that the ratio of capital to labor remains unchanged, enterprises expand the scale of production and input capital, which increases the demand for labor in the same proportion and creates a large number of employment opportunities. Second, technological innovation has given birth to new products and new industries. With the development of technological innovation, science and technology are more and more able to meet the needs of consumers, which is matched by the emergence of new industries and new products, which not only absorb the new labor force but also attract the labor force from traditional and basic sectors.

In summary, the employment effect of technological innovation has a dual nature: technological innovation improves labor productivity and organic composition of capital by changing production and operation, reduces the demand for labor by enterprise production and operation, and thus forms the "substitution effect" of capital on labor. At the same time, technological innovation has the effect of improving labor productivity and reducing production cost. On this basis, expanding the production scale will increase the enterprises' demand for labor, and create new jobs by developing new products and opening up new production and service fields, thus forming the "compensation effect" of technological innovation on employment. Focusing on the relationship between technological innovation and employment, and the first hypothesis is stated as follows:

**Hypothesis 1.** *Technological innovation has a positive impact on enterprise employment.*

*2.2. Environmental Regulation and Enterprise Employment*

Environmental policy tools mainly include two categories: one is the regulation policy (regulating, for example, the total carbon emissions of the enterprise), and the other is the carbon tax (although there is no aggregate constraint, it will increase the production cost of the enterprise). In fact, as early as the 1970s, many developed countries began to implement strict environmental regulation for different pollution problems. This has also led to a series of problems, for example, people are worried that environmental regulation will increase the production cost of enterprises, weaken the competitive advantage and production scale of enterprises, and reduce the number of workers they can absorb, which will result in a country-wide unemployment problem. One of the focuses is on the potential

negative employment effects of environmental regulation. In 1990, a study published by the American Business Association showed that the Clean Air Act Amendments caused losses of between 1 million and 2 million jobs. Considering the consequences of the loss of employment, the implementer of the Act used a fund of $5 million per year to subsidize the unemployed workers (Goodstein, 1996) [41]. Morgenstern et al. [42] also mentioned that in the 1990 poll, 1/3 of the respondents reported that their work was threatened by environmental regulation. However, many studies have found that a large number of closures and firings caused by environmental regulation are "exaggerated" because people tend to ignore the new employment opportunities brought by environmental protection (Goodstein, 1994) [43]. In fact, economic theory does not provide a clear answer to the question of whether stronger environmental standards will lead to fewer jobs.

There are three main viewpoints on the relationship between environmental regulation and domestic and overseas employment. First, environmental regulation reduces the number of jobs. Dissou and Sun [44] used the general equilibrium framework to analyze the impact of carbon emission reduction policy on labor demand. The results show that the carbon emission reduction policy has a negative impact on employment when the licensed income is transferred to the family. Second, environmental regulation has increased the number of jobs. Mishra and Smyth [45] argued that when environmental regulation acts on industries with high labor demand, such as the environmental protection industry and related service industry, it will increase labor demand and improve employment level. Gray et al. [46] utilized the DID method to evaluate the employment effect of the U.S. EPA regulations in the pulp and paper industry, and found that this policy does not significantly reduce employment, but may slightly increase employment. Zhao [47] estimated the impact of investment intensity of industrial pollution control on regional employment by using a simultaneous equation model, and empirically found that investment in pollution control effectively promotes employment growth. Chen [48] and Zhang [49] also reached similar conclusions. Third, the impact of the environmental regulation on the number of jobs is uncertain or shows a U-shaped relationship. Kahn and Mansur [18] insisted that the spatial transfer of employment caused by differences in environmental regulatory standards will have an uncertain impact on employment in different regions. Yan et al. [50] used the panel threshold model to verify the difference in the impact of environmental regulation on employment with industrial structure and environmental regulation were taken as threshold variables. The results show that low-level environmental regulation cannot damage employment, and the key to achieving a win-win situation between environmental regulation and employment is to increase the proportion of the tertiary industry. On the basis of Morgenstern's theoretical framework, Wang et al. [51] introduced industry characteristic parameters and used panel data of 38 industrial industries in China to verify the impact of environmental regulation on employment in different industries. They found that there is a U-shaped relationship between them, which would promote employment when environmental regulation crosses the threshold. Based on the heterogeneity of labor income and education level, Li [52] tested the impact and difference of environmental regulation on employment through the model of the factors affecting employment under the equilibrium of producers, and it was found that there is a U-shaped dynamic relationship between environmental regulation and employment. Li [53] divided the employment structure according to the pollution degree and technical level, and found that the heterogeneity of the industry resulted in significant differences in the shape and position of the U-shaped curve.

To sum up, the impact of environmental regulation on employment includes a negative scale effect and an uncertain substitution effect. Therefore, the second hypothesis is stated as follows:

**Hypothesis 2.** *Environmental regulation has a positive impact on enterprise employment.*

*2.3. The Regulatory Role of Environmental Regulation*

Confronted with environmental regulation, if enterprises choose to use cleaner production technology, the relationship between employment creation and employment loss generated by

production technology itself will become more uncertain. Clean technology is usually divided into two categories: terminal decontamination technology and process improvement technology. Terminal pollution control technology is mainly aimed at the pollutants produced in the production process. These pollution control technologies need to increase labor input in the operation and monitoring process, thus creating some employment opportunities. At the same time, these terminal technologies may also transform the by-products (such as residues) produced in the production process into commodities, thus increasing the profits of enterprises and corresponding employment opportunities. Process improvement technology mainly affects the entire production process of the enterprise and can directly affect the technological progress of the enterprise (for example, new equipment installed will produce fewer emissions in the operation process). Process improvement techniques will undoubtedly reduce the need for productive workers. This is because: compared with backward technology, advanced technology usually requires less labor input, which will produce the substitution effect of technology [54]. At this point, it is entirely up to the nature of clean technology itself to determine whether pollution control activities are "alternative" or "complementary" to the needs of the workforce.

Faced with environmental regulation, enterprises may improve production technology, including terminal emission reduction technology (pollution-control technical progress) and production process technology (production-oriented technical progress), in order to meet the standards of clean production. The progress of production technology will, on the one hand, reduce the production costs and product prices of manufacturers, which will increase the demand for products and expand the scale of production, resulting in increase the demand for labor; on the other hand, it may bring about an increase in production efficiency and a decrease in labor demand per unit of output, resulting in the decline of social employment. In other words, the impact of technological innovation on employment under the influence of environmental regulation is uncertain [55,56]. Therefore, it is an urgent issue for us to clarify the impact of technological innovation on enterprise employment under the influence of government environmental regulation. Based on the above analysis, the third hypothesis is stated as follows:

**Hypothesis 3.** *Environmental regulation plays a negative regulatory role in the impact of technological innovation on enterprise employment.*

In summary, the theoretical model of hypotheses constructed in this paper is shown in Figure 1 below.

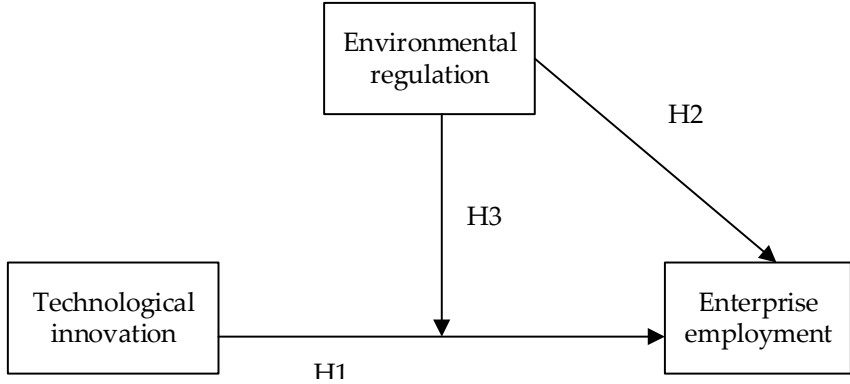

**Figure 1.** The theoretical model of hypotheses H1–H3.

## 3. Material and Methods

### 3.1. Model Specification

In order to test the research hypothesis proposed in the previous paper, three basic panel data models were established according to the research of existing literature models and the actual

situation for this paper. The general form of the econometric model for the exploration of the impact of technological innovation on enterprise employment is provided by Equation (1), while that of environmental regulation on enterprise employment is provide by Equation (2). On the basis of Equations (1) and (2), Equation (3) introduces the interaction term between environmental regulation and technological innovation to test the moderating effect of environmental regulation on the relationship between technological innovation and enterprise employment. Based on the above theoretical analysis and the influencing factors of employment in manufacturing enterprises, the following econometric models are used in this paper:

$$\ln \text{employ}_{it} = \phi_0 + \phi_1 \ln \text{r\&d}_{it} + \phi_2 \ln \text{rev}_{it} + \phi_3 \ln \text{size}_{it} + \phi_4 \ln \text{wage}_{it} + \phi_5 \text{age}_{it} + \delta_{it} \tag{1}$$

$$\ln \text{employ}_{it} = \varphi_0 + \varphi_1 \ln \text{r\&d}_{it} + \varphi_2 \ln \text{pace}_{it} + \varphi_3 \ln \text{rev}_{it} + \varphi_4 \ln \text{size}_{it} + \varphi_5 \ln \text{wage}_{it} + \varphi_6 \text{age}_{it} + \varepsilon_{it} \tag{2}$$

$$\ln \text{employ}_{it} = \eta_0 + \eta_1 \ln \text{r\&d}_{it} + \eta_2 \ln \text{pace}_{it} + \eta_3 \ln \text{r\&d}_{it} \times \ln \text{pace}_{it} + \eta_4 \ln \text{rev}_{it}$$
$$+ \eta_5 \ln \text{size}_{it} + \eta_6 \ln \text{wage}_{it} + \eta_7 \text{age}_{it} + \mu_{it} \tag{3}$$

In the above models, the subscripts i and t represent the enterprise and the year, respectively, $\phi$, $\varphi$, $\eta$ represent the intercept terms, and $\delta$, $\varepsilon$, $\mu$ represent the random disturbance terms. Among them, "employ" is the number of listed manufacturing enterprises, "r&d" indicates the R&D investment of listed companies, "pace" indicates the environmental protection expenditure of listed companies, "rev" indicates the indicators of operation revenue, "size" indicates the size of the company, "wage" indicates the salary level of the enterprise, and "age" indicates the number of years a company has been listed.

### 3.2. Sample Selection and Data Source

This paper takes A-share manufacturing companies listed on the Shanghai and Shenzhen stock exchanges from 2011 to 2017 as the research objects. In order to eliminate the adverse impact of abnormal samples on the empirical results, this paper screened and sorted out the original data according to the following standards: (1) excluding the listed companies with incomplete data, (2) excluding the listed companies of ST* and ST (It refers to the special treatment of listed companies with abnormal financial status or other conditions.), and (3) excluding the nature of foreign ownership and other listed companies. After screening, 124 companies were finally obtained, with a total of 868 research samples.

The financial data used in this paper are taken from the China Securities Market & Accounting Research (CSMAR) database (http://www.gtarsc.com/Home) and the Wind database (https://www.wind.com.cn/newsite/edb.html). The data of environmental expenditures include pollution treatment expenditures, green environmental protection fees, sewage charges, etc. disclosed in the management fees in the notes of the annual report of the CSMAR database, which were obtained through manual sorting. The data for R&D investment was mainly disclosed in the R&D investment amount of listed companies by CSMAR. The employment of manufacturing enterprises comes from the number of employees in the comprehensive information document of listed company governance.

The screening process of sample companies is shown in Table 1 below.

**Table 1.** Sample screening process.

| Criteria | Companies | Observations |
|---|---|---|
| Screen A-share listed manufacturing companies | 2354 | 16,478 |
| Exclude companies that do not disclose environmental expenditure | 2155 | 15,085 |
| Exclude companies with abnormal financial conditions | 10 | 70 |
| Exclude companies with incomplete environmental expenditur | 5 | 35 |
| Exclude companies that do not disclose R&D data | 6 | 42 |
| Exclude companies with abnormal financial conditions | 4 | 28 |
| Exclude companies with incomplete R&D data | 35 | 245 |
| Exclude companies with missing number of employees data | 2 | 14 |
| Exclude foreign and other companies | 13 | 91 |
| The remaining sample companies | 124 | 868 |

### 3.3. Variable Definitions and Descriptions

#### 3.3.1. Dependent Variable

Enterprise employment ("employ"). Employment is the foundation of people's livelihood and the source of wealth, which plays a very important role in the process of social development. Manufacturing in China has a very important strategic significance for the development of the country. At present, China's manufacturing employment is facing unprecedented challenges and opportunities. This paper adopts the number of employees in the comprehensive information file of listed company governance to represent the number of enterprise employees.

#### 3.3.2. Independent Variables

Environmental regulation ("pace"). In the face of increasingly tight environmental regulation, enterprises are subject to various environmental protection indicators and will increase spending on environmental protection, such as increased sewage charges, green fees, and so on. This paper selects the total environmental protection expenditure related to enterprise pollution reduction and emission control expenditures to measure the environmental regulation of enterprises.

Technological innovation ("r&d"). When enterprises are undergoing transformation and upgrading, it is indispensable to increase R&D investment to improve their core competitiveness. This paper draws on the viewpoint of Li et al. (2018) [57], using the R&D investment of enterprises as an indicator to measure the ability of technological innovation. The more R&D investment funds an enterprise has, the stronger the technological innovation capability of enterprises.

#### 3.3.3. Control Variables

Operating income ("rev"). Operating income is an important financial index of an enterprise that reflects the economic benefits of the enterprise and relates to the operating status of the enterprise. The enterprise's operating conditions will reflect the impact of the enterprise's labor demand to some extent.

Enterprise size ("size"). Enterprise scale is the division of enterprise production and business scope and is one of the basic factors affecting enterprise employment. The larger the enterprise is, the larger the labor demand is. In this paper, the natural logarithm of the total assets of a company is used to measure the scale of an enterprise.

Wage level ("wage"). The salary level is an important factor affecting the employment behavior of enterprises. In this paper, the cash paid to employees and the cash paid for employees in the cash flow statement disclosed in the financial statements of enterprises are used to calculate the wage level. The calculation formula is as follows: wage level = the cash paid to employees + the cash paid for employees/number of employees

Enterprise age ("age"). As the age of the listing increases, the growth rate of the company usually slows down, which in turn affects the labor demand of the company. Therefore, the company's listing age is chosen to control the impact of the company's time to market on employment. The calculation formula is as follows: enterprise age = the sample year − the year the company was listed.

Details of the utilized variables are outlined in Table 2. The average employment value of manufacturing enterprises was 8.25, and the standard deviation was relatively large at 1.11, indicating that there are significant differences in employment among different enterprises. The standard deviation of R&D investment was 1.61, while the standard deviation of environmental protection expenditure was 1.72. According to Table 2, it can be found that during the sample study period, the dispersion degree of each variable was relatively high, indicating that there was significant heterogeneity among enterprises.

**Table 2.** Descriptive statistics of variables.

| Variable | Obs | Mean | SD | Minimum | Maximum |
|---|---|---|---|---|---|
| ln(employ) | 868 | 8.25 | 1.11 | 5.56 | 11.53 |
| ln(r&d) | 868 | 17.79 | 1.61 | 12.83 | 21.78 |
| ln(pace) | 868 | 15.00 | 1.72 | 8.55 | 19.00 |
| ln(r&d) × ln(pace) | 868 | 267.86 | 46.83 | 147.31 | 406.95 |
| ln(rev) | 868 | 3.64 | 1.44 | 0.48 | 7.50 |
| ln(size) | 868 | 4.11 | 1.24 | 1.55 | 7.60 |
| ln(wage) | 868 | 11.28 | 0.42 | 10.20 | 13.24 |
| age | 868 | 16.68 | 4.01 | 6 | 31 |

## 4. Empirical Results and Discussion

Based on the panel data of A-share manufacturing companies listed on the Shanghai and Shenzhen stock exchanges from 2011 to 2017 as a sample for the econometric test, the Hausman test results indicated that the fixed effect model should be selected by considering the heteroscedasticity and cross-section correlation using the "xtscc, fe" command to perform regression to reduce the effects of heteroscedasticity and cross-section correlation on regression results. Due to the large differences in product characteristics and production processes between enterprises in different manufacturing industries, industries with different levels of pollution and different levels of technology respond differently to environmental regulation. Therefore, we classified manufacturing enterprises according to the difference between pollution degree [58] and technical level [53] in industries to which the enterprises belong (see Appendix A), and further examined the relationship between environmental regulation, technological innovation, and enterprise employment on the basis of classification.

### 4.1. Full-Sample Regression

The results of the regression of all the samples of 124 listed companies are shown in Table 3. Model 1 is the return of technological innovation to employment, model 2 is the return of environmental regulation to employment, and model 3 is the return of the interaction item between technological innovation and environmental regulation on employment.

**Table 3.** The results of the full-sample regression.

| Model | (1) | (2) | (3) |
|---|---|---|---|
| Variable | ln(employ) | ln(employ) | ln(employ) |
| ln(r&d) | 0.0150 * | 0.0130 | 0.333 *** |
| | (1.80) | (1.58) | (5.47) |
| ln(rev) | 0.304*** | 0.297 *** | 0.288 *** |
| | (11.09) | (10.97) | (10.81) |
| ln(size) | 0.313 *** | 0.292 *** | 0.278 *** |
| | (9.72) | (9.07) | (8.79) |
| ln(wage) | −0.734 *** | −0.734 *** | −0.742 *** |
| | (−24.53) | (−24.87) | (−25.56) |
| age | 0.0373 *** | 0.0373 *** | 0.0399 *** |
| | (8.08) | (8.19) | (8.86) |
| ln(pace) | | 0.0426 *** | 0.412 *** |
| | | (4.62) | (5.87) |
| ln(r&d) × ln(pace) | | | −0.0207 *** |
| | | | (−5.31) |
| _cons | 13.25 *** | 12.76 *** | 7.224 *** |
| | (41.39) | (38.30) | (6.61) |
| N | 868 | 868 | 868 |

*t* statistics in parentheses, * *p* < 0.1, ** *p* < 0.05, *** *p* < 0.01.

According to Table 3, model 1 shows that technological innovation was significantly positively correlated with enterprise employment, with a coefficient of 0.0150. Hypothesis 1 was verified, and the increase in R&D investment of enterprises did not reduce the labor demand of enterprises. From the enterprise level, the impact of technological innovation on enterprise employment was positive, which may be because enterprise R&D investment improved the productivity, increased the demand for products, expanded the scale of enterprises, and won a greater market share. The compensation effect was greater than the substitution effect, and the demand for labor increased. Meanwhile, model 2 shows that there was a positive correlation between enterprise environmental regulation and labor demand, with a significant coefficient of 0.0426, and hypothesis 2 was verified. This indicates that the labor demand of enterprises rose with the increase of environmental protection expenditure, which supports the hypothesis of a double dividend of employment. When the environmental protection expenditure increased, although the cost of the enterprise became larger, which was due to environmental protection activities, such as environmental treatment at the end of production, the cost effectiveness was less than the substitution effect, and it did not crowd out the employment. Moreover, model 3 shows that enterprise environmental regulation played an obvious regulatory role in technological innovation and enterprise employment, with a negative direction and a coefficient of −0.0207, with hypothesis 3 being verified. That is, with the increase of environmental protection expenditure, the marginal effect of R&D investment on enterprise employment decreased. This may be because the environmental protection expenditure of enterprises squeezed the cost of R&D investment, which led to a decrease in the employment effect of R&D investment. It may also be that the progress of pollution control technology caused by environmental regulation was less than the production-oriented technological progress, or the progress of pollution control technology was biased toward the use of capital rather than labor, which caused the employment effect of technological innovation to have diminishing returns.

For the control variables, operation income, enterprise size, and years of listing had a positive impact on enterprise employment, and enterprise wage level had a negative impact on enterprise employment. The higher the operating income of an enterprise was, the higher the possibility of obtaining a higher net profit was, and the enterprise could increase the output of products and input more labor force. At the same time, the larger the enterprise scale was, the larger the fixed asset investment was, and the more labor input that was needed for the normal operation of the enterprise. The longer a company was listed, the more research and development costs were recovered, and the market share of the company was gradually expanded, providing more employment opportunities. The higher the wage level was, the higher the labor cost would be, and most enterprises chose to reduce the number of employees and reduce the loss.

The results of the full-sample indicate that: (1) Technological innovation had a direct positive effect on enterprise employment, that is, technological innovation promoted employment increase. (2) Environmental regulation had positive effects on enterprises, and it did not cause a reduction in employment in enterprises, which verified the hypothesis of double dividend between environmental regulation and employment. (3) Environmental regulation had a significant negative effect on the regulatory effect of technological innovation and enterprise employment. The continuous improvement of the level of environmental regulation stimulated the technological upgrading of enterprises to a certain extent. The upgrading of technology was accompanied by the use of large and efficient mechanical equipment, which led to the substitution effect of mechanical equipment on the labor force and led to a decrease in the number of employees. According to the above theoretical analysis, at present, technological progress stimulated by China's environmental regulation is more reflected in the progress of production technology, thus reducing employment, while the progress of pollution control technology is not obvious, so the development of the environmental protection industry brought about by the progress of pollution control technology and the growth of employment have not been highlighted [59–61].

### 4.2. Comparison of Different Ownership Structures

This section further explores the differences in the impacts of environmental regulation and technological innovation on the employment of manufacturing enterprises for different types of enterprises. Enterprises with different ownership structures often face different levels of government intervention. The government will intervene in the employment behavior of state-owned enterprises. The government's intervention costs for private enterprises are relatively high, which prevents redundant employment in private enterprises [62]. At the same time, different relations between government and enterprises also have a certain impact on enterprises' R&D investment [63]. Under the environmental policy, enterprises of different natures face different levels of intervention, so it is necessary to conduct a comparative analysis of enterprises with different ownership structures.

The results of models 4–9 in Table 4 show that there were significant differences in the impact of environmental regulation and technological innovation on enterprise employment in enterprises with different property rights. According to Table 4, models 4–6 are regression results of state-owned enterprises, and models 7–9 are the empirical results of private enterprises. In state-owned enterprises, model 4 indicates that the influence coefficient of enterprise technological innovation on enterprise employment was 0.0285, model 5 indicates that the influence coefficient of environmental regulation on enterprise employment was 0.0473, and model 6 shows that the influence coefficient of interaction between environmental regulation and technological innovation on enterprise employment was −0.0256. In private enterprises, the influence coefficient of technological innovation on enterprise employment in model 7 was 0.0286, model 8 indicates that the influence coefficient of environmental regulation on enterprise employment was 0.0361, and model 9 indicates that the influence coefficient of interaction between environmental regulation and technological innovation on enterprise employment was −0.00992, but there was no significant influence. It can be seen that compared with private enterprises, technological innovation of state-owned enterprises had a slightly smaller positive effect on enterprise employment, and environmental regulation had a more significant positive effect on enterprise employment [64]. Meanwhile, environmental regulation had a greater negative regulatory effect on the relationship between technological innovation and enterprise employment.

**Table 4.** Regression results of enterprises with different ownership structures.

| | State-Owned Enterprises | | | Private Enterprises | | |
|---|---|---|---|---|---|---|
| Model | (4) | (5) | (6) | (7) | (8) | (9) |
| Variable | ln(employ) | ln(employ) | ln(employ) | ln(employ) | ln(employ) | ln(employ) |
| ln(r&d) | 0.0285 *** | 0.0256 ** | 0.428 *** | 0.0286 ** | 0.0281 ** | 0.174 * |
| | (2.73) | (2.46) | (5.34) | (1.99) | (1.99) | (1.93) |
| ln(rev) | 0.250 *** | 0.244 *** | 0.231 *** | 0.328 *** | 0.322 *** | 0.320 *** |
| | (7.13) | (7.02) | (6.82) | (7.74) | (7.72) | (7.70) |
| ln(size) | 0.277 *** | 0.252 *** | 0.232 *** | 0.321 *** | 0.304 *** | 0.297 *** |
| | (6.49) | (5.84) | (5.52) | (6.85) | (6.57) | (6.41) |
| ln(wage) | −0.685 *** | −0.694 *** | −0.716 *** | −0.821 *** | −0.816 *** | −0.817 *** |
| | (−14.21) | (−14.50) | (−15.35) | (−23.55) | (−23.77) | (−23.84) |
| age | 0.0205 *** | 0.0215 *** | 0.0246 *** | 0.0604 *** | 0.0592 *** | 0.0607 *** |
| | (3.34) | (3.54) | (4.16) | (9.10) | (9.07) | (9.23) |
| ln(pace) | | 0.0473 *** | 0.506 *** | | 0.0361 *** | 0.212 * |
| | | (3.04) | (5.51) | | (3.52) | (1.96) |
| ln(r&d) × ln(pace) | | | −0.0256 *** | | | −0.00992 |
| | | | (−5.06) | | | (−1.64) |
| _cons | 13.31 *** | 12.86 *** | 6.055 *** | 13.31 *** | 12.83 *** | 10.25 *** |
| | (26.08) | (24.38) | (4.21) | (33.12) | (30.69) | (6.30) |
| N | 476 | 476 | 476 | 392 | 392 | 392 |

*t* statistics in parentheses, * $p < 0.1$, ** $p < 0.05$, *** $p < 0.01$.

In conclusion, in both state-owned enterprises and private enterprises, technological innovation and environmental regulation had a positive impact on enterprise employment. This may be because private enterprises were less negatively regulated by environmental regulation, so the positive employment effect of technological innovation of private enterprises was larger than that of state-owned enterprises in general. It is worth noting that in private enterprises, the negative regulatory effect of environmental regulation on the relationship between technological innovation and employment was not significant.

In order to intuitively demonstrate this regulatory effect and further compare the regulatory effect of environmental regulation between state-owned enterprises and private enterprises, we drew the regulatory effect diagram corresponding to enterprises with different property rights, as shown in Figure 2. Both in private enterprises and state-owned enterprises, the regulatory effect of environmental regulation was negative, but we found that the slope of state-owned enterprises was smaller than that of private enterprises, which means that for state-owned enterprises, environmental regulation had a greater negative impact on the employment effect of technological innovation.

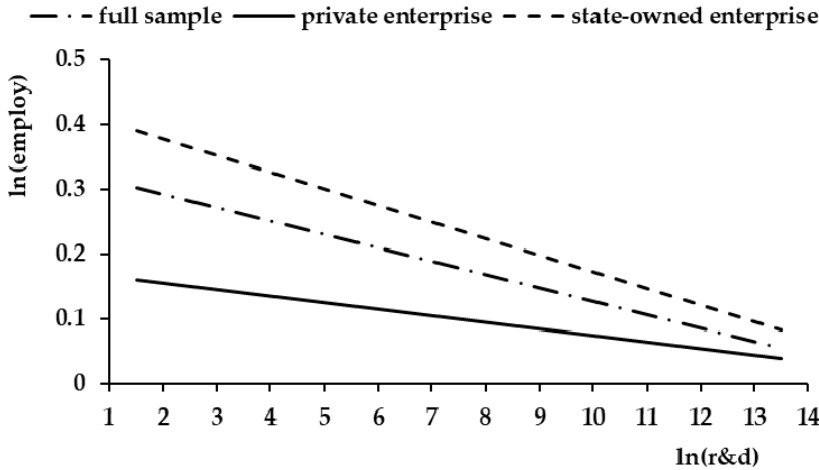

**Figure 2.** Comparison of the regulatory role of enterprises with different ownership structures.

### 4.3. Comparison of Different Industry Characteristics

Since the relationship between environmental regulation, technological innovation, and enterprise employment may also be heterogeneous with the degree of pollution and the level of technology, manufacturing enterprises were classified according to pollution level and technical level of the industries they belong to. First, we divided manufacturing enterprises into clean enterprises and pollution-intensive enterprises. Second, we divided manufacturing enterprises into high-tech enterprises and low- and medium-tech enterprises.

Due to the difference in the degree of pollution among different industries, the impact of technological innovation on the employment of enterprises under environmental policies may also have different results. Therefore, we analyzed the employment situation of enterprises in different industries according to the degree of pollution. The results of models 10–15 in Table 5 show that there were significant differences in the impact of environmental regulation and technological innovations on employment of enterprises with different levels of pollution. According to Table 5, models 10–12 are regression results of enterprises in the clean industries, and models 13–15 are empirical results of enterprises in the pollution-intensive industries. For enterprises in the clean industries, model 10 shows that the influence coefficient of technological innovation on enterprise employment was 0.0788, model 11 shows that the influence coefficient of environmental regulation on enterprise employment was 0.0568, and model 12 shows that the influence coefficient of interaction between environmental regulation and technological innovation on enterprise employment was −0.00221, but it was not significant. For enterprises in pollution-intensive industries, the influence coefficient of

enterprise technological innovation on enterprise employment in model 13 was 0.00685, the influence coefficient of environmental regulation on enterprise employment in model 14 was 0.0334, and the influence coefficient of interaction between environmental regulation and technological innovation on enterprise employment in model 15 was −0.0285. Compared with enterprises in pollution-intensive industries, technological innovation of enterprises in the clean industries had a greater positive effect on enterprise employment, environmental regulation had a relatively significant positive effect on enterprise employment [53], and environmental regulation had a smaller negative regulatory effect on the relationship between technological innovation and enterprise employment.

In summary, both the technological innovation of enterprises in clean industries and that of enterprises in pollution-intensive industries had a positive impact on employment, and environmental regulation played a positive role in the employment performance of enterprises. The negative regulation of environmental regulation of enterprises in clean industries was smaller than that in pollution-intensive industries, and eventually, positive employment effect of technological innovation in clean industries was greater than that in pollution-intensive industries. The regulation of environmental regulation in the clean industries was smaller, and the employment growth of innovative enterprises in the clean industries was in line with the green employment demand under the consensus of the international community, which was conducive to improving the green employment capacity.

**Table 5.** Regression results of listed enterprises in industries with different pollution levels.

| | Enterprises of Clean Industry | | | Enterprises of Pollution-Intensive Industry | | |
|---|---|---|---|---|---|---|
| **Model** | **(10)** | **(11)** | **(12)** | **(13)** | **(14)** | **(15)** |
| **Variable** | **ln(employ)** | **ln(employ)** | **ln(employ)** | **ln(employ)** | **ln(employ)** | **ln(employ)** |
| ln(r&d) | 0.0788 *** | 0.0765 *** | 0.108 | 0.00685 | 0.00477 | 0.449 *** |
| | (3.40) | (3.36) | (1.03) | (0.80) | (0.56) | (5.99) |
| ln(rev) | 0.214 *** | 0.213 *** | 0.212 *** | 0.325 *** | 0.318 *** | 0.314 *** |
| | (3.92) | (3.96) | (3.93) | (10.54) | (10.38) | (10.58) |
| ln(size) | 0.307 *** | 0.287 *** | 0.289 *** | 0.268 *** | 0.249 *** | 0.226 *** |
| | (5.23) | (4.94) | (4.94) | (6.83) | (6.34) | (5.92) |
| ln(wage) | −0.815 *** | −0.819 *** | −0.820 *** | −0.695 *** | −0.693 *** | −0.698 *** |
| | (−15.02) | (−15.35) | (−15.30) | (−19.67) | (−19.78) | (−20.62) |
| age | 0.0553 *** | 0.0507 *** | 0.0507 *** | 0.0320 *** | 0.0330 *** | 0.0369 *** |
| | (5.98) | (5.50) | (5.50) | (6.19) | (6.44) | (7.39) |
| ln(pace) | | 0.0568 *** | 0.0963 | | 0.0334 *** | 0.539 *** |
| | | (3.10) | (0.74) | | (3.19) | (6.31) |
| ln(r&d) × ln(pace) | | | −0.00221 | | | −0.0285 *** |
| | | | (−0.31) | | | (−5.96) |
| _cons | 13.12 *** | 12.57 *** | 12.01 *** | 13.10 *** | 12.68 *** | 4.933 *** |
| | (21.41) | (19.97) | (6.30) | (35.49) | (32.64) | (3.65) |
| N | 301 | 301 | 301 | 567 | 567 | 567 |

*t* statistics in parentheses, * $p < 0.1$, ** $p < 0.05$, *** $p < 0.01$.

Similarly, in order to further compare the regulatory effect of environmental regulation between enterprises in the clean industries and those in the pollution-intensive industries, we drew the regulatory effect diagram corresponding to enterprises in industries with different pollution levels, as shown in Figure 3. Both in enterprises of clean industries and pollution-intensive industries, the regulatory effect of environmental regulation was negative, but we found that the slope of pollution-intensive industries was less than that of clean industries, which means for enterprises of pollution-intensive industries, the environmental regulation effect on employment of technological innovation negative influence was greater.

innovation negative influence was greater.

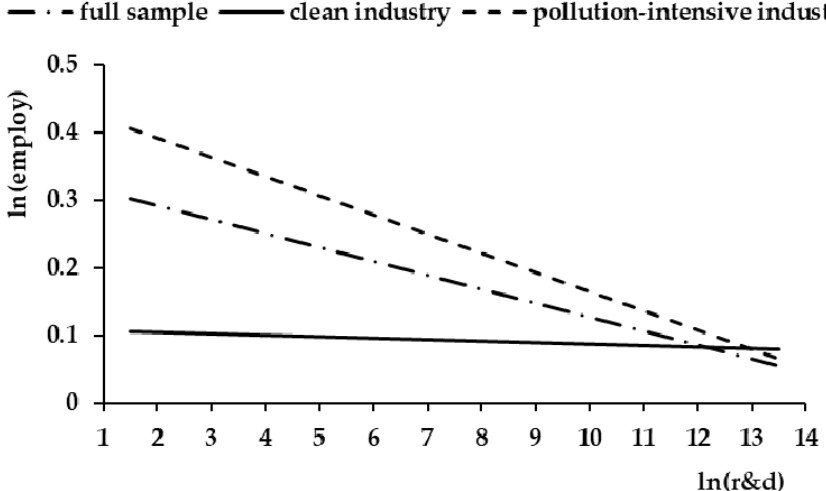

**Figure 3.** Comparison of regulatory roles of industries with different levels of pollution.

Next, we analyzed the situation of enterprises in industries with different technical levels. The results of models 16–21 in Table 6 show that there were differences in the regulating effects of environmental regulation and technological innovation on enterprise employment in industries with different technological levels.

**Table 6.** Regression results of listed enterprises in industries with different technical levels.

| | Enterprises of High-Tech Industry | | | Enterprises of Low- and Medium-Tech Industry | | |
|---|---|---|---|---|---|---|
| **Model** | **(16)** | **(17)** | **(18)** | **(19)** | **(20)** | **(21)** |
| **Variable** | ln(employ) | ln(employ) | ln(employ) | ln(employ) | ln(employ) | ln(employ) |
| ln(r&d) | 0.0305 ** | 0.0294 ** | 0.249 *** | 0.00622 | 0.00432 | 0.373 *** |
| | (2.20) | (2.16) | (3.01) | (0.65) | (0.45) | (4.32) |
| ln(rev) | 0.227 *** | 0.225 *** | 0.227 *** | 0.349 *** | 0.345 *** | 0.325 *** |
| | (5.80) | (5.82) | (5.93) | (9.52) | (9.44) | (9.06) |
| ln(size) | 0.343 *** | 0.327 *** | 0.317 *** | 0.204 *** | 0.185 *** | 0.183 *** |
| | (8.23) | (7.93) | (7.71) | (4.21) | (3.76) | (3.83) |
| ln(wage) | −0.823 *** | −0.820 *** | −0.825 *** | −0.603 *** | −0.602 *** | −0.606 *** |
| | (−21.76) | (−22.02) | (−22.29) | (−12.92) | (−12.93) | (−13.39) |
| age | 0.0581 *** | 0.0542 *** | 0.0558 *** | 0.0205 *** | 0.0223 *** | 0.0248 *** |
| | (8.69) | (8.14) | (8.40) | (3.42) | (3.68) | (4.20) |
| ln(pace) | | 0.0485 *** | 0.310 *** | | 0.0242 * | 0.441 *** |
| | | (3.79) | (3.17) | | (1.90) | (4.51) |
| ln(r&d) × ln(pace) | | | −0.0146 *** | | | −0.0235 *** |
| | | | (−2.69) | | | (−4.29) |
| _cons | 13.74 *** | 13.15 *** | 9.313 *** | 12.57 *** | 12.28 *** | 5.874 *** |
| | (33.13) | (30.11) | (6.25) | (25.94) | (24.25) | (3.74) |
| N | 497 | 497 | 497 | 371 | 371 | 371 |

*t* statistics in parentheses, * $p < 0.1$, ** $p < 0.05$, *** $p < 0.01$.

According to Table 6, models 16–18 are regression results of enterprises in high-tech industries, and models 19–21 are empirical results of enterprises in low- and medium-tech industries. In high-tech enterprises, model 16 shows that the impact coefficient of technological innovation on enterprise employment was 0.0305, model 17 shows that the impact coefficient of environmental regulation on enterprise employment was 0.0485, model 18 shows that the interaction term of environmental regulation and technological innovation on enterprise employment was −0.0146, and the regulatory effect was significantly negative. For enterprises in the low- and medium-tech industries, the influence coefficient of technological innovation on enterprise employment in model 19 was 0.00622, the

influence coefficient of environmental regulation on enterprise employment in model 20 was 0.0242, and the influence coefficient of interaction between environmental regulation and technological innovation on enterprise employment in model 21 was −0.0235. The negative regulation effect of environmental regulation was relatively large and significant in both high-tech and low- and medium-tech industries, which means that under the influence of environmental regulation, the employment effect of technological innovation had diminishing marginal returns. Compared with enterprises in low- and medium-tech industries, the technological innovation of enterprises in high-tech industries had a greater positive effect on enterprise employment, environmental regulation had a positive effect on enterprise employment, and environmental regulation had a smaller negative regulating effect on the relationship between R&D investment and enterprise employment.

Above all, both technological innovation of enterprises in high-tech industries and that of low- and medium-tech industries had a significant positive effect on enterprise employment. Environmental regulation played a positive role in enterprise employment. The negative regulatory effect of environmental regulation of enterprises in high-tech industries was smaller than that in the low- and medium-tech industries, and the positive employment effect of the technological innovation of the enterprises of high-tech industries was greater than that of the low- and medium-tech enterprises. The environmental regulation of high-tech enterprises had a less negative effect on the innovation employment effect. The employment growth of high-tech enterprises was in line with the basic requirements of China's current development of a high-quality economy, which was conducive to optimizing the employment structure and improving the quality of employment. This conclusion is consistent with the creation effect of technological innovation in high-tech sectors derived by Piva and Vivarelli [40].

To further compare the regulating effect of environmental regulation between enterprises in high-tech industries and enterprises in low- and medium-tech industries, we drew the regulating effect diagram corresponding to enterprises in industries with different technical levels, as shown in Figure 4. Both in high-tech industries and low- and medium-tech industries, environmental regulation of regulating effect was negative, but we found that the slope of the low- and medium-tech industries was smaller than that of high-tech industries, which means that for enterprises of low- and medium-tech industries, the environmental regulation effect on employment of technological innovation negative influence was greater.

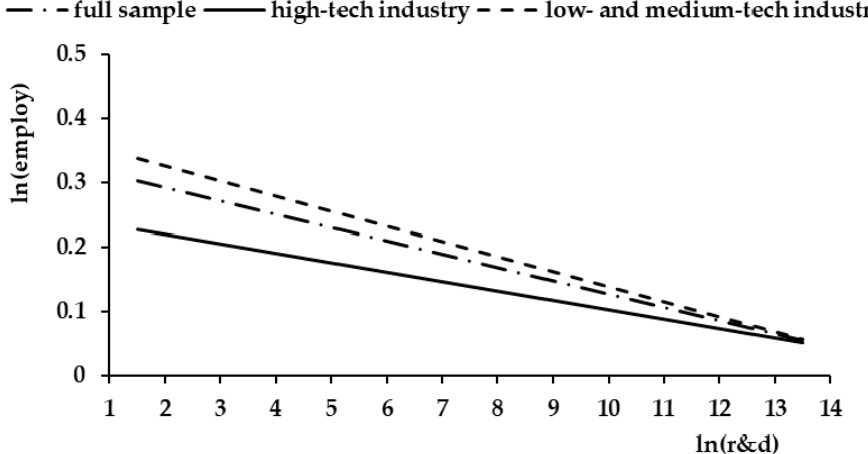

**Figure 4.** Comparison of the regulatory role of enterprises in different technical levels.

### 4.4. Robustness Test

In order to prove the stability of the research conclusions, this paper used the GMM-SYS method to test the stability of the relationship between environmental regulation, technological innovation, and employment growth. Due to the paucity of observations, when the entire sample was split in

sub-samples, the GMM method could not be continued. According to Table 7, the results of the total sample analysis show that the direct impact of technological innovation on employment growth was a positive effect (0.359), the regression coefficient of environmental regulation on employment growth was significantly positive (0.414), and the regulatory effect of environmental regulation on technological innovation and employment growth was negative (−0.0214). The direction and magnitude of the regression coefficients of each model were basically consistent with the above, indicating that research conclusions of this paper are relatively stable.

**Table 7.** Regression results of robustness test.

| Model | (1) | (2) | (3) |
|---|---|---|---|
| **Variable** | **ln(employ)** | **ln(employ)** | **ln(employ)** |
| ln(employ) | 0.675 *** | 0.639 *** | 0.563 *** |
| | (5.53) | (5.15) | (4.66) |
| ln(r&d) | 0.0508 ** | 0.0473 ** | 0.359 * |
| | (2.31) | (2.15) | (1.89) |
| ln(rev) | 0.0565 | 0.0497 | 0.0737 ** |
| | (1.35) | (1.30) | (2.17) |
| ln(size) | 0.171 ** | 0.168 ** | 0.244 *** |
| | (2.07) | (2.08) | (2.89) |
| ln(wage) | −0.331 *** | −0.342 *** | −0.360 *** |
| | (-3.94) | (−4.54) | (−4.77) |
| age | 0.00429 | 0.00665 | 0.00477 |
| | (1.11) | (1.63) | (1.04) |
| ln(pace) | | 0.0340 ** | 0.414 * |
| | | (2.37) | (1.71) |
| ln(r&d) × ln(pace) | | | −0.0214 * |
| | | | (−1.67) |
| _cons | 4.560 *** | 4.538 *** | −0.510 |
| | (2.88) | (3.16) | (−0.17) |
| N | 744 | 744 | 744 |

*t* statistics in parentheses, * $p < 0.1$, ** $p < 0.05$, *** $p < 0.01$.

## 5. Conclusions and Policy Implications

### 5.1. Conclusions

This paper considered the research question: "In the context of environmental protection and high-quality economic development, does environmental regulation hinder the employment creation of technological innovation in enterprises?" To do so, panel data of listed Chinese manufacturing companies (2011–2017) were selected and empirical tests were carried out by adopting the moderating effect model. The following conclusions are drawn:

(1) The overall impact of technological innovation on enterprise employment was reflected in the creation effect. The employment effect of technological innovation of state-owned enterprises was slightly smaller than that of private enterprises. The employment effect of technological innovation of enterprises in the clean industries was larger than that of enterprises in the pollution-intensive industries. The employment effect of technological innovation of enterprises in the high-tech industries was larger than that of enterprises in the low- and medium-tech industries.

(2) On the whole, the direct impact of environmental regulation on enterprise employment was significantly positive. The positive effect of environmental regulation on the employment in state-owned enterprises was greater than that in private enterprises. The coefficient of positive correlation between environmental regulation and employment of enterprises in the clean industries was larger than that of enterprises in the pollution-intensive industries. The positive effect of environmental regulation on the employment of enterprises in the high-tech industries was larger than that of enterprises in the low- and medium-tech industries.

(3) The regulatory effect of environmental regulation on the relationship between technological innovation and enterprise employment was negative, and there was obvious enterprise heterogeneity. Among them, the negative regulation impact of environmental regulation on the employment effect of technological innovation in state-owned enterprises was larger than that in private enterprises. The environmental regulation of enterprises in pollution-intensive industries had a negative adjustment effect on the relationship between technological innovation and employment, which was more than that in clean industries. Environmental regulation of low- and medium-tech industries had a bigger negative influence than that of high-tech industries. At the present stage in China, the development of pollution-control technology is relatively lagging behind, which leads to the insufficient impetus for the development of the environmental protection industry and a limited increase in employment. However, environmental regulation stimulates the progress of production technology more obviously, which leads to the more prominent phenomenon of reducing employment.

*5.2. Policy Implications*

The important policy implications of these conclusions are as follows:

(1) Increase government environmental research and development subsidies to improve enterprises' ability regarding environmental technology innovation. Both environmental regulation and technological innovation are positive for the employment of enterprises. It may be that the environmental protection expenditure of enterprises increases the environmental cost of enterprises, and the investment in research and development of enterprises may be somewhat squeezed out. Therefore, under the regulation of environmental regulation, the employment growth margin of technological innovation is diminishing. By increasing enterprises' investment in environmental research and development and improving enterprises' ability of environmental innovation, the tension between environmental protection expenditure and research and development investment can be turned into coordinated development so as to achieve a win-win situation of high-quality development and employment.

(2) Support the green development of private enterprises, and cultivate and strengthen private leading enterprises in environmental protection. Compared with state-owned enterprises, environmental regulation of private enterprises has a less negative regulating effect on the relationship between technological innovation and employment. Private enterprises have an innovation consciousness toward a constantly deepening innovation of practical technology; can quickly adapt to the market; have flexible management mechanisms; can constantly increase investment in facilities and research and development in the field of ecological and environmental governance; and actively explore collaborative governance, industrial integration, and other mode innovation, which is the new force of ecological and environmental governance.

(3) Eliminate enterprises with high energy consumption and high pollution, and encourage green transformation and technological upgrading. The negative moderating effect of environmental regulation on the employment effect of technological innovation of enterprises in pollution-intensive industries is relatively more obvious than that of enterprises in clean industries. Meanwhile, the crowding out effect of environmental regulation on the employment effect of technological innovation of enterprises low- and medium-tech industries is larger than that of high-tech industries. Accelerating the transformation of heavy polluting enterprises into clean enterprises and low- and medium-tech enterprises into high-tech enterprises can weaken the negative impact of environmental regulation on the employment effect of technological innovation of enterprises.

*5.3. Limitations*

Although this study provides valuable insights, it has limitations, which should serve to stimulate further research. First, because the data of pollution emission of enterprises are not available, this paper selected the enterprise environmental protection expenditure as the index to measure the intensity of environmental regulation, which has some defects. Second, from the

theoretical level, the technological innovation induced by environmental regulation can be divided into production-oriented technological innovation and pollution-control technological innovation. In this paper, we have not further differentiated the types of technological innovation in the empirical research. It is conducive to deeply analyzing the transmission path that environmental regulation affects the relationship between technological innovation and employment through figuring out whether the technological innovation induced by environmental regulation is production-oriented or pollution-control technological innovation. In further research, we will try to expand the research by taking panel data from China's manufacturing industries as the research subject to improve the above issues.

**Author Contributions:** Conceptualization, D.L. and J.Z.; Methodology, D.L.; Investigation, D.L.; Writing—Original Draft Preparation, D.L.; Writing—Review and Editing, D.L. and J.Z. All authors contributed to writing the paper.

**Funding:** The authors are grateful to the financial support provided by the National Natural Science Foundation of China (71373198), the National Social Science Foundation of China (11BJY043).

**Acknowledgments:** The research data of this work is serviced by the China Securities Market & Accounting Research (CSMAR) database and the Wind database.

**Conflicts of Interest:** The authors declare no conflict of interest.

## Appendix A

1. Classification results according to pollution degree: Clean industries (17): Extraction of Petroleum and Natural Gas; Manufacture of Tobacco; Manufacture of Textile Wearing and Apparel; Manufacture of Leather, Fur, Feather and Related Products and Footware; Processing of Timber, Manufacture of Wood, Bamboo, Rattan, Palm, and Straw Products; Manufacture of Furniture; Printing, Reproduction of Recording Media; Manufacture of Articles for Culture, Education and Sport Activity; Manufacture of Medicines; Manufacture of Rubber and Plastic; Manufacture of Metal Products; Manufacture of General Purpose Machinery; Manufacture of Special Purpose Machinery; Manufacture of Transport Equipment; Manufacture of Electrical Machinery and Equipment; Manufacture of Communication Equipment, Computers and Other Electronic Equipment; Manufacture of Measuring Instrument. Pollution-intensive industries (17): Mining and Washing of Coal; Mining and Processing of Ferrous Metal Ores; Mining and Processing of Non-ferrous Metal Ores; Mining and Processing of Non-metal Ores; Processing of Food from Agricultural Products; Manufacture of Foods; Manufacture of Wine, Drinks and Refined Tea; Manufacture of Textile; Manufacture of Paper and Paper Products; Processing of Petroleum, Coking,

Processing of Nuclear Fuel; Manufacture of Raw Chemical Materials and Chemical Products; Manufacture of Chemical Fibers; Manufacture of Non-metallic Mineral Products; Smelting and Pressing of Ferrous Metals; Smelting and Pressing of Non-ferrous Metals; Production and Supply of Electric Power and Heat Power; Production and Supply of Gas.

2. Classification results according to technical level: High-tech industries (10): Manufacture of Raw Chemical Materials and Chemical Products; Manufacture of Medicines; Manufacture of Chemical Fibers; Manufacture of Metal Products; Manufacture of General Purpose Machinery; Manufacture of Special Purpose Machinery; Manufacture of Transport Equipment; Manufacture of Electrical Machinery and Equipment; Manufacture of Computers, Communication, and Other Electronic Equipment; Manufacture of Measuring Instrument. Low- and medium-tech industries (24): Mining and Washing of Coal; Extraction of Petroleum and Natural Gas; Mining and Processing of Ferrous Metal Ores; Mining and Processing of Non-ferrous Metal Ores; Mining and Processing of Non-metal Ores; Processing of Food from Agricultural Products; Manufacture of Foods; Manufacture of Wine, Drinks and Refined Tea; Manufacture of Tobacco; Manufacture of Textile; Manufacture of Textile Wearing and Apparel; Manufacture of Leather, Fur, Feather and Related Products and Footware; Processing of Timber, Manufacture of Wood, Bamboo, Rattan, Palm, and Straw Products; Manufacture of Furniture; Manufacture of Paper and Paper Products; Printing, Reproduction of Recording Media; Manufacture of Articles for Culture, Education and Sport Activity; Processing of Petroleum, Coking,

Processing of Nuclear Fuel; Manufacture of Rubber and Plastic; Manufacture of Non-metallic Mineral Products; Smelting and Pressing of Ferrous Metals; Smelting and Pressing of Non-ferrous Metals; Production and Supply of Electric Power and Heat Power; Production and Supply of Gas.

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
