# Peer review of "The Role of Environmental Regulation and Technological Innovation in the Employment of Manufacturing Enterprises: Evidence from China"

_sustainability, doi:10.3390/su11102982_

Round 1

Reviewer 1 Report

Related to the established literature on the impact of technology upon employment, this paper takes on board a rather un-investigated issue, namely the role of environmental regulations both as an additional employment driver and in its interaction with the new technologies. While very well fitting the Sustainability aims and scope and obtaining novel and interesting results, this version is affected by important shortcomings that have to be dealt with in the revision.

·        The location of the paper (Section 1) should be enriched by the discussion of some recent influential papers, at the moment neglected by the Authors, namely:

-Frey C.B., Osborne M.A. (2017). The future of employment: how susceptible are jobs to computerisation? Technological Forecasting and Social Change, 114, pp.254-280.

-Van Roy, V. - Vertesy, D. - Vivarelli, M. (2018), Technology and Employment: Mass Unemployment or Job Creation? Empirical Evidence from European Patenting Firms, Research Policy*, 47, 1762-1776.

·        Section 2 should be substantially extended and updated, since the section does not give justice to a huge and complex empirical economic literature on the subject; at least the following works should be discussed and added to the reference list:

-Aldieri, L., Vinci, C. P.  (2018), Innovation effects on employment in high-tech and low-tech industries: evidence from large international firms within the triad, Eurasian Business Review, 8, 229-243.

-Bogliacino F., Pianta M. (2010). Innovation and employment. A reinvestigation using revised Pavitt classes. Research Policy, 39, pp.799-809.

-Bogliacino, F. - Vivarelli, M. (2012), The Job Creation Effect of R&D Expenditures, Australian Economic Papers, 51, 96-113.

-Bogliacino, F., Piva, M., Vivarelli, M., 2012. R&D and employment: An application of the LSDVC estimator using European data. Economics Letters, 116, 383-404.

-Evangelista, R., Vezzani, A., 2012. The impact of technological and organizational innovations on employment in European firms. Industrial and corporate change 21, 871-899.

-Falk, M., Hagsten, E. (2018) Employment impacts of market novelty sales: evidence for nine European Countries,  Eurasian Business Review, 8, 119–137.

-Freeman C., Soete L. (1994). Work for All or Mass Unemployment? Computerised Technical Change into the Twenty-first Century. London-New York: Pinter.

- Gagliardi, L. & Marin, G. & Miriello, C. (2016). The greener the better? Job creation effects of environmentally-friendly technological change, Industrial and Corporate Change, 25, 779-807.

-Katsoulacos Y.S. (1984). Product innovation and employment. European Economic Review, 26, pp.83-108.

-Piva, M. - Vivarelli, M. (2018), Technological Change and Employment: Is Europe Ready for the Challenge?, Eurasian Business Review, 8, 13-32.

-Vivarelli, M. (1995), The  Economics  of  Technology  and Employment: Theory and Empirical Evidence, Elgar, Cheltenham.

·        The econometric specification is adequate, but two important qualifications are necessary: 1) in principle, a dynamic specifications should be preferable (that is with the inclusion of the lagged dependent variable as an additional regressor): here the paucity of observations prevents from pursuing this methodology (GMM type), especially when the entire sample is split in sub-samples; however, this important limitation should be admitted, qualified and properly discussed; 2) the proxy for technology (R&D) is OK, but - as discussed in the literature listed above - R&D expenditures are much more correlated with the labour-friendly product innovation rather than with the labour-saving process innovation: this ex-ante optimistic bias should be admitted and discussed.

·        The theoretical section 5.4 is unrelated to the previous empirical analysis and not necessary. Since it is at odds with the format of the paper and not essential, it should be dropped all together.

Minor points:

line 37: international division of labour

line 45: qualify what is the government work report of the two sessions

line 65: capital organic composition is a Marxian terminology, please translate it into capital intensity

line 81 and related reference: Van Reenen is the correct surname

line 82: Piva &Vivarelli is misspelled in the references (Piva and not Pisa)

line 86: qualify what intended for HJMP model (reference 3, that should be completed: please avoid et al. in the references).

line 102: Jordi model?

line 131: for this paper, however our study is

line 182: to new products and new industries

line 268: qualify what intended for ST* and ST

Author Response

Response to Reviewer 1 Comments

Dear Reviewers:

Thank you for comments concerning our manuscript entitled The Role of Environmental Regulation and Technological Innovation in the Employment of Manufacturing EnterprisesEvidence from China (ID: 501390). Those comments are all valuable and very helpful for revising and improving our paper, as well as the important guiding significance to our researches. We have studied comments carefully and have made correction which we hope meet with approval. Revised portion are marked in red in the paper. The main corrections in the paper and the responds to the reviewer’s comments are as flowing:

Point 1: The location of the paper (Section 1) should be enriched by the discussion of some recent influential papers, at the moment neglected by the Authors, namely:

-Frey C.B., Osborne M.A. (2017). The future of employment: how susceptible are jobs to computerisation? Technological Forecasting and Social Change, 114, pp.254-280.

-Van Roy, V. - Vertesy, D. - Vivarelli, M. (2018), Technology and Employment: Mass Unemployment or Job Creation? Empirical Evidence from European Patenting Firms, Research Policy*, 47, 1762-1776.

Response 1: We have enriched the Section 1 by discussion of some influential papers in order to demonstrate that the subject is truly known and to clearly highlight what are the intended objectives and what is the real contribution that is been made.

Point 2:    Section 2 should be substantially extended and updated, since the section does not give justice to a huge and complex empirical economic literature on the subject; at least the following works should be discussed and added to the reference list:

-Aldieri, L., Vinci, C. P.  (2018), Innovation effects on employment in high-tech and low-tech industries: evidence from large international firms within the triad, Eurasian Business Review, 8, 229-243.

-Bogliacino F., Pianta M. (2010). Innovation and employment. A reinvestigation using revised Pavitt classes. Research Policy, 39, pp.799-809.

-Bogliacino, F. Vivarelli, M. (2012), The Job Creation Effect of R&D Expenditures, Australian Economic Papers, 51, 96-113.

-Bogliacino, F., Piva, M., Vivarelli, M., 2012. R&D and employment: An application of the LSDVC estimator using European data. Economics Letters, 116, 383-404.

-Evangelista, R., Vezzani, A., 2012. The impact of technological and organizational innovations on employment in European firms. Industrial and corporate change 21, 871-899.

-Falk, M., Hagsten, E. (2018) Employment impacts of market novelty sales: evidence for nine European Countries, Eurasian Business Review, 8, 119–137.

-Freeman C., Soete L. (1994). Work for All or Mass Unemployment? Computerised Technical Change into the Twenty-first Century. London-New York: Pinter.

- Gagliardi, L. & Marin, G. & Miriello, C. (2016). The greener the better? Job creation effects of environmentally-friendly technological change, Industrial and Corporate Change, 25, 779-807.

-Katsoulacos Y.S. (1984). Product innovation and employment. European Economic Review, 26, pp.83-108.

-Piva, M. Vivarelli, M. (2018), Technological Change and Employment: Is Europe Ready for the Challenge? Eurasian Business Review, 8, 13-32.

-Vivarelli, M. (1995), The Economics of Technology and Employment: Theory and Empirical Evidence, Elgar, Cheltenham.

Response 2: Section 2 has been substantially extended and updated by enough bibliographical citations that try to substantiate the statements made and give support to the hypotheses of the paper as well as give justice to a huge and complex empirical economic literature on the subject; much more than the mentioned above works have been discussed and added to the reference list.

Point 3: The econometric specification is adequate, but two important qualifications are necessary: 1) in principle, a dynamic specifications should be preferable (that is with the inclusion of the lagged dependent variable as an additional regressor): here the paucity of observations prevents from pursuing this methodology (GMM type), especially when the entire sample is split in sub-samples; however, this important limitation should be admitted, qualified and properly discussed; 2) the proxy for technology (R&D) is OK, but - as discussed in the literature listed above - R&D expenditures are much more correlated with the labour-friendly product innovation rather than with the labour-saving process innovation: this ex-ante optimistic bias should be admitted and discussed.

Response 3: 1) In order to prove the stability of the research conclusions, this paper uses the GMM-SYS method to test the stability of the relationship among environmental regulation, technological innovation and employment growth. Due to the paucity of observations, when the entire sample is split in sub-samples, the GMM method cannot be continued. 2) R&D expenditures are much more correlated with the labour-friendly product innovation rather than with the labour-saving process innovation: this ex-ante optimistic bias has been admitted and discussed.

Point 4: The theoretical section 5.4 is unrelated to the previous empirical analysis and not necessary. Since it is at odds with the format of the paper and not essential, it should be dropped all together.

Response 4: The theoretical section 5.4 is dropped all together.

Point 5: Minor points:

line 37: international division of labour

line 45: qualify what is the government work report of the two sessions

line 65: capital organic composition is a Marxian terminology, please translate it into capital intensity

line 81 and related reference: Van Reenen is the correct surname

line 82: Piva &Vivarelli is misspelled in the references (Piva and not Pisa)

line 86: qualify what intended for HJMP model (reference 3, that should be completed: please avoid et al. in the references).

line 102: Jordi model?

line 131: for this paper, however our study is

line 182: to new products and new industries

line 268: qualify what intended for ST* and ST

Response 5:

line 37: “international division” is corrected as “international division of labour”.

line 45: The government work report of the two sessions has been qualified as a report delivered by the premier of the state council to the National People's Congress and submitted to deputies of the National People's Congress for deliberation of the NPC&CPPCC.

line 65: “capital organic composition” has been translated into “capital intensity”.

line 81 The surname “Reenen” is corrected as “Van Reenen” and “Reenen J V” is corrected as “Van Reenen, J” in the related reference.

line 82: “Pisa” is corrected “Piva” in the references.

line 86: HJMP model has been qualified as a model framework based on production functions; reference 3 is completed and other references are checked.

line 102: Jordi model is correct presentation which is mentioned in reference 31.

line 131: “however” is added.

line 182: “to new industries and created more jobs” is corrected as “to new products and new industries”

line 268: ST* and ST is qualified as the enterprise that appears financial condition or other unusual condition.

We tried our best to improve the manuscript and made some changes in the manuscript. These changes will not influence the content and framework of the paper. And here we did not list the changes but marked in red in revised paper.

We appreciate for reviewers’ warm work earnestly, and hope that the correction will meet with approval.

Once again, thank you very much for your comments and suggestions.

Reviewer 2 Report

In general, the article deals with an interesting topic, although in our opinion it has important flaws.

The main flaw is that it has many similarities with the resolution of an exercise, but does not expose the main foundations of the literature on innovation and environmental regulation that would sustain many of the claims made therein. For example:

a) The introduction of any scientific article usually exposes the main contributions made previously, in order to demonstrate that the subject is truly known, and to clearly highlight what are the intended objectives and what is the real contribution that is been made. In this case, there is no bibliographical reference throughout the introduction.

b) There are no enough bibliographical citations that try to substantiate the statements made and give support to the hypotheses of the paper. Without intending to be exhaustive, hypothesis 1 is not supported by any bibliographical citation. It seems that there has been no scientific literature that has previously treated this hypothesis. In addition, the bibliographical foundations for the hypotheses 2 and 3 approach are very scarce, practically nonexistent.

c) In the section "Empirical Results and Discussion", there is a mere description of the results obtained. Discussing the results requires comparing the results obtained with those obtained by other previous empirical investigations, in order to discuss the similarities and differences, and, consequently, to justify from the perspective of the existing theoretical and empirical literature the relevance and application of the results of the paper. From my perspective, the authors should try to expand their bibliographical sources very considerably, since a broad contribution exists in the international literature on this topic. In addition, the contribution of section "5.4. Further Discussion" is far from being a discussion of the results obtained, and, in a certain sense, does not seem to be a necessary and appropriate inclusion.

In short, the theoretical foundations and the discussion of the results require a lot more work. It is necessary to read and understand previous international contributions that have been made on this topic. There is little previous literature cited, and most only of Chinese origin.

There are many other deficiencies that can be highlighted, but as long as the indicated deficiencies are not corrected, any other correction does not make sense.

Author Response

Response to Reviewer 2 Comments

Dear Reviewers:

Thank you for comments concerning our manuscript entitled The Role of Environmental Regulation and Technological Innovation in the Employment of Manufacturing EnterprisesEvidence from China (ID: 501390). Those comments are all valuable and very helpful for revising and improving our paper, as well as the important guiding significance to our researches. We have studied comments carefully and have made correction which we hope meet with approval. Revised portion are marked in red in the paper. The main corrections in the paper and the responds to the reviewer’s comments are as flowing:

Point 1: The introduction of any scientific article usually exposes the main contributions made previously, in order to demonstrate that the subject is truly known, and to clearly highlight what are the intended objectives and what is the real contribution that is been made. In this case, there is no bibliographical reference throughout the introduction.

Response 1: Numerous bibliographical references have been added to the introduction. Section1 is enriched by the discussion of some recent influential papers.

Point 2: There are no enough bibliographical citations that try to substantiate the statements made and give support to the hypotheses of the paper. Without intending to be exhaustive, hypothesis 1 is not supported by any bibliographical citation. It seems that there has been no scientific literature that has previously treated this hypothesis. In addition, the bibliographical foundations for the hypotheses 2 and 3 approach are very scarce, practically non-existent.

Response 2: Section 2 is substantially extended and updated, and give justice to a huge and complex empirical economic literature on the subject. A large number of references have been listed to support the hypotheses of the paper in Section2.

Point 3: In the section "Empirical Results and Discussion", there is a mere description of the results obtained. Discussing the results requires comparing the results obtained with those obtained by other previous empirical investigations, in order to discuss the similarities and differences, and, consequently, to justify from the perspective of the existing theoretical and empirical literature the relevance and application of the results of the paper. From my perspective, the authors should try to expand their bibliographical sources very considerably, since a broad contribution exists in the international literature on this topic. In addition, the contribution of section "5.4. Further Discussion" is far from being a discussion of the results obtained, and, in a certain sense, does not seem to be a necessary and appropriate inclusion.

Response 3: We compare the results obtained with those obtained by other previous empirical investigations. The theoretical section 5.4 is dropped all together.

We tried our best to improve the manuscript and made some changes in the manuscript. These changes will not influence the content and framework of the paper. And here we did not list the changes but marked in red in revised paper.

We appreciate for reviewers’ warm work earnestly, and hope that the correction will meet with approval.

Once again, thank you very much for your comments and suggestions.

Reviewer 3 Report

The article provides interesting insights into study of the role of environmental regulation in the impact of the technological innovation on enterprise development.

In general, the structure of the article is well presented however I would suggest some changes and improvements. In order to keep the logical and consistent structure, the content of the article should be simplified and clearly presented. The following changes would be reasonable:

·        to combine section “2. Literature review” with section “3. Theoretical Framework and Research Hypotheses” (Lines: 77 and 152).

·        change the subsection title “5.4. Further Discussion” into the section title “5. Discussion” (Line: 535)

Moreover, the content concerning the research aim etc. presented in lines 131-151 must be moved into Introduction section.

The introduction should provide not only a research background and the briefly description of the content of each section of the paper but also an the indication of the main objective, methods used as well as explanation of intended contribution.

Moreover it is difficult to assess the significance of the research background because in the introduction there are no one literature reference. This must be complemented.  

It is also necessary to include the precise research objective in the Abstract.

The general proofreading is necessary because of some small shortcomings that should be improved, e.g. in line 264 the word “listed” is used two times.

Author Response

Response to Reviewer 3 Comments

Dear Reviewers:

Thank you for comments concerning our manuscript entitled The Role of Environmental Regulation and Technological Innovation in the Employment of Manufacturing EnterprisesEvidence from China (ID: 501390). Those comments are all valuable and very helpful for revising and improving our paper, as well as the important guiding significance to our researches. We have studied comments carefully and have made correction which we hope meet with approval. Revised portion are marked in red in the paper. The main corrections in the paper and the responds to the reviewer’s comments are as flowing:

Point 1: to combine section “2. Literature review” with section “3. Theoretical Framework and Research Hypotheses” (Lines: 77 and 152).

Response 1: We have combined Section2 with Section3.

Point 2: change the subsection title “5.4. Further Discussion” into the section title “5. Discussion” (Line: 535).

Response 2: The theoretical section 5.4 is dropped all together.

Point 3: Moreover, the content concerning the research aim etc. presented in lines 131-151 must be moved into Introduction section.

Response 3: The content concerning the research aim etc. presented in lines 131-151 is moved into Introduction section.

Point 4: The introduction should provide not only a research background and the briefly description of the content of each section of the paper but also a the indication of the main objective, methods used as well as explanation of intended contribution. Moreover, it is difficult to assess the significance of the research background because in the introduction there are no one literature reference. This must be complemented.   It is also necessary to include the precise research objective in the Abstract.

Response 4: We have enriched the Section 1 by discussion of some influential papers in order to demonstrate that the subject is truly known and to clearly highlight what are the intended objectives and what is the real contribution that is been made.

Point 5: The general proofreading is necessary because of some small shortcomings that should be improved, e.g. in line 264 the word “listed” is used two times.

Response 5: One of the words “listed” is deleted and other parts of the paper are checked.

We tried our best to improve the manuscript and made some changes in the manuscript. These changes will not influence the content and framework of the paper. And here we did not list the changes but marked in red in revised paper.

We appreciate for reviewers’ warm work earnestly, and hope that the correction will meet with approval.

Once again, thank you very much for your comments and suggestions.

Round 2

Reviewer 2 Report

With the incorporation of bibliographic references in the text, the paper has improved. However, I believe that the following issues should be addressed:

1) In line 185 it is indicated: "and that technological innovation has a greater effect on employment than product innovation"

This comment is contradictory, since product innovation is a technological innovation (product innovation and process innovation).

2) In lines 194-195 it is indicated: "Wu [31] adopted Jordi model".

The reader does not have to know what the Jordi Model is. It is necessary to enter a new reference

3) On table 1 the following expression appears twice: “Exclude companies with incomplete R&D data”. We assume that it is an error. If so, it must be corrected.

4) Appendix A does not mention the references of where the Classification pollution degree, Pollution-intensive industries, and technical level come from. There are different classifications, so it is necessary that the authors indicate which one they use.

5) In the "References" section there is no uniform pattern. Sometimes before indicating the pages of an article a "pp." is written, while most of the time it is not.  In our opinion, one of the two reference criteria should prevail. For example:

3. Frey, C.B.; Osborne, M.A. The future of employment: How susceptible are jobs to computerisation? Technological Forecasting and Social Change[J]. 2017,114, pp.254-280.

23. Van Reenen, J. Employment and technological innovation: Evidence from U. K. manufacturing firms[J]. Journal of Labor Economics,1997,15(2): 255-284.

We have found numerous differences of another type, too many to list here. Authors should try to homogenize the way of quoting.

Author Response

Dear Reviewer:

Thank you for comments concerning our manuscript entitled The Role of Environmental Regulation and Technological Innovation in the Employment of Manufacturing EnterprisesEvidence from China (ID: 501390). Those comments are all valuable and very helpful for revising and improving our paper, as well as the important guiding significance to our researches. We have studied comments carefully and have made correction which we hope meet with approval. Revised portion are marked in red in the paper. The main corrections in the paper and the responds to the reviewer’s comments are as flowing:

Point 1: In line 185 it is indicated: "and that technological innovation has a greater effect on employment than product innovation" This comment is contradictory, since product innovation is a technological innovation (product innovation and process innovation).

Response 1: "and that technological innovation has a greater effect on employment than product innovation" is corrected as "and that process innovation has a greater effect on employment than product innovation".

Point 2: In lines 194-195 it is indicated: "Wu [31] adopted Jordi model".  The reader does not have to know what the Jordi Model is. It is necessary to enter a new reference.

Response 2: “Wu [31] adopted Jordi model of the correlation mechanism between technological innovation and employment to investigate the employment creation effect of different types of innovation in enterprises” is corrected as “Wu [31] investigated the employment creation effect of different types of innovation in enterprises”.

Point 3: On table 1 the following expression appears twice: “Exclude companies with incomplete R&D data”. We assume that it is an error. If so, it must be corrected.

Response 3: This is two different sample screening process, the first is to remove samples that do not disclose R&D data, and the second is to remove samples with incomplete R&D data. To make a distinction, first “Exclude companies with incomplete R&D data” is corrected as “Exclude companies that don't disclose R&D data.

Point 4: Appendix A does not mention the references of where the Classification pollution degree, Pollution-intensive industries, and technical level come from. There are different classifications, so it is necessary that the authors indicate which one they use.

Response 4: Therefore, we classify manufacturing enterprises according to the difference between pollution degree [58] and technical level [53] in industries to which the enterprises belong.

Point 5: In the "References" section there is no uniform pattern. Sometimes before indicating the pages of an article a "pp." is written, while most of the time it is not.  In our opinion, one of the two reference criteria should prevail. For example:  3. Frey, C.B.; Osborne, M.A. The future of employment: How susceptible are jobs to computerisation? Technological Forecasting and Social Change[J]. 2017,114, pp.254-280.  23. Van Reenen, J. Employment and technological innovation: Evidence from U. K. manufacturing firms[J]. Journal of Labor Economics,1997,15(2): 255-284.  We have found numerous differences of another type, too many to list here. Authors should try to homogenize the way of quoting.

Response 5: We've tried to standardize the references according to “23. Van Reenen, J. Employment and technological innovation: Evidence from U. K. manufacturing firms[J]. Journal of Labor Economics,1997,15(2): 255-284.”.

We tried our best to improve the manuscript and made some changes in the manuscript. These changes will not influence the content and framework of the paper. And here we did not list the changes but marked in red in revised paper.

We appreciate for reviewer’ warm work earnestly, and hope that the correction will meet with approval.

Once again, thank you very much for your comments and suggestions.
